# Statistical meta-analysis to investigate the association between the Interleukin-6 (IL-6) gene polymorphisms and cancer risk

**Md. Harun-Or-Roshid[1], Md. Borqat Ali[1], Jesmin[2]\*, Md. Nurul Haque Mollah [1]\***

**1** Bioinformatics Laboratory, Department of Statistics, University of Rajshahi, Rajshahi, Bangladesh,
**2** Department of Genetic Engineering and Biotechnology, University of Dhaka, Dhaka, Bangladesh

\* mollah.stat.bio@ru.ac.bd (MNHM); jesmin@univdhaka.edu (J)

**Data Availability Statement:** The meta-data were collected from published papers by online searching. All data used in this paper and their

## Abstract

A good number of genome-wide association studies (GWAS), including meta-analyses, reported that single nucleotide polymorphisms (SNPs) of the IL-6 gene are significantly associated with various types of cancer risks, though some other studies reported insignificant association with cancers, in the literature. These contradictory results may be due to variations in sample sizes and/or deficiency of statistical modeling. Therefore, an attempt is made to provide a more comprehensive understanding of the association between the IL-6 gene SNPs (rs1800795, rs1800796, rs1800797) and different cancer risks, giving the weight on a large sample size, including different cancer types and appropriate statistical modeling with the meta-dataset. In order to attain a more reliable consensus decision about the association between the IL-6 gene polymorphisms and different cancer risks, in this study, we performed a multi-case statistical meta-analysis based on the collected information of 118 GWAS studies comprising of 50053 cases and 65204 control samples. Results from this Meta-analysis indicated a significant association ($p$-value < 0.05) of the IL-6 gene rs1800796 polymorphism with an overall increased cancer risk. The subgroup analysis data based on cancer types exhibited significant association ($p$-value < 0.05) of the rs1800795 polymorphism with an overall increased risk of cervical, liver and prostate cancers; the rs1800796 polymorphism with lung, prostate and stomach cancers; and the rs1800797 polymorphism with cervical cancer. The subgroup analysis of ethnicity data showed a significant association ($p$-value < 0.05) of an overall cancer risk with the rs1800795 polymorphism for the African and Asian populations, the rs1800796 polymorphism for the Asian only and the rs1800797 polymorphism in the African population. Comparative discussion showed that our multi-case meta-analyses received more support than any previously reported individual meta-analysis about the association between the IL-6 gene polymorphisms and cancer risks. Results from this study, more confidently showed that the IL-6 gene SNPs (rs1800795, rs1800796 and rs1800797) in humans are associated with increased cancer risks. Therefore, these three polymorphisms of the IL-6 gene have the potential to be evaluated as a population based rapid, low-cost PCR prognostic biomarkers for different types of cancers diagnosis and research.

Reference ID are given in the paper and Supporting Information files.

**Funding:** This work was supported by Rajshahi University Research Project (A-1416/5/52/RU/Science-11/18-19)

**Competing interests:** The authors have declared that no competing interests exist.

## Introduction

Cancer is a leading cause of death worldwide. According to the World Health Organization (WHO), 9.6 million deaths occurred in 2018 from 18.1 million cancer patients all over the globe. It has been estimated that the incidence of cancer occurrences might be increased by 50% to 15 million new cases by the year 2020 [1]. The GLOBOCAN database published the extent of mortality and outbreak in 2018 from 36 types of cancer in 185 countries [2]. According to the recent literature reviews, it is very much evident now that cancer is a multi-factorial progressive disorder that developed under the influence of genes and their interactions [2–4].

Interleukin-6 (IL-6) gene encodes a cytokine that functions in inflammation and has been reported in association with cancers in the literature for many years [3, 4]. Growing evidence suggests an important role for pro-inflammatory cytokines like the IL-6 gene in the microenvironment of tumor development and regarded as an important tumor promoting factor in various types of human cancers including breast, oral, gastrointestinal, prostate, and, colorectal cancer [5–12]. The IL-6 rs1800795 (-174G/C) polymorphism is a significant predictor for susceptibility of prostate cancer and bone metastasis in northwest Iranian population [11]. A number of meta-analysis of the IL-6 gene polymorphisms with cancer risk were conducted based on small sample sizes [123–130]. Some GWA studies have also reported that some polymorphisms of the IL-6 gene are insignificantly associated with blood cancer [13–25] and significantly associated with breast cancer [26–36] whereas some other studies [37–40] reported no association with breast cancer. Similarly, for cervical and colon cancer development, some studies reported a significant association [41–46 and 49–56 respectively] whereas other studies reported no significant association [44, 47, 48, 57–61 respectively]. The same scenario persists for liver cancer: associated [62–68] and not "associated [69–71]; lung cancer: associated [72–77] and not associated [78–83]; prostate cancer: associated [11, 84–91] and not associated [92–98]; stomach cancer: claimed a significant association [106–112] and some other studies claimed insignificant association [111, 113–115]. Also, some GWA studies investigated the association of the IL-6 gene polymorphisms with thyroid cancer [116, 117], ovarian cancer [118], pancreatic cancer [119], neuroblastoma [120, 121] and renal cell carcinoma [122]. Thus we observed from the above discussion that different types of cancers were influenced by the three SNPs (rs1800795, rs1800796 and rs1800797) of the IL-6 gene. We also observed that the reported results varied across studies and therefore, remain inconclusive, which may be occurred due to the smaller sample size and different ethnic populations.

To overcome the ambiguity of GWAS findings, some Author's performed meta-analysis based on only one of three important SNPs (rs1800795, rs1800796 and rs1800797) of the IL-6 gene or only one type of cancer to take more reliable and valid conclusion [123–129]. It should be mentioned here that a meta-analysis is conducted by the complete coverage of all relevant studies, solving the heterogeneity problem, and exploring the robustness of main findings using sensitivity analysis. Those meta-analysis reported that (i) the rs1800795 polymorphism of the IL-6 gene shows significant association with cervical [123] and colorectal [124] cancers, but insignificant association with stomach cancer [128, 129], (ii) thers1800796 polymorphism shows contradictory association with stomach cancer [111] and insignificant association with lung cancer [126, 127] and (iii) thers1800797 polymorphism shows insignificant association with colorectal cancer [124], stomach cancer [129] and all type of risks [128]. Thus those meta-analysis reports on the IL-6 gene were not consistent in their common issues. Zhou et al. [131] performed multi-case meta-analysis considering all of three important SNPs of the IL-6 gene as mentioned previously, three different ethnicities (Asian, African, Caucasian), nine types of cancers based on 49,408 cancers and 61,790 control cases. They reported that the IL-6 gene is significantly associated with the overall cancer risk. Particularly, they reported

significant association of IL-6 gene with 2 types of cancer risks (liver and prostate) and insignificant association with 7 types of cancer risks (breast, cervical, colorectal, gastric, lung, lymphoma and myeloma) by the sub-group analysis of cancer types. Obviously, their specific report [130] contradicts with the results of other single-case meta-analyses [123–129] in the cases of their common interest, which may be happened due to smaller sample sizes, ethnicity and the deficiency of statistical modeling with the meta-dataset. For example, none of those meta-analyses [123–130] checked the model adequacy by the goodness of fit test. To estimate the combined effects, all of them used fixed effect (FE) or random effect (RE) models corresponding to the homogeneity or heterogeneity of effects which cannot give the guarantee of model adequacy [133].Therefore, in this paper, an attempt was made to provide a more comprehensive understanding about the association between the IL-6 gene SNPs (rs1800795, rs1800796, rs1800797) and different types of cancer risks, giving the weight on large sample size, more cancer types and appropriate statistical modeling based on the goodness of fit test [133] with the multi-case meta-dataset.

## Materials and methods

### Search strategy

Text mined data of the competent articles retrieved from PubMed, PubMed Central, Google Scholar, Web of Science and other online literature databases, published up to February 2019 in the English language were only considered for this meta-analysis. The following keywords were used for searching: (i) IL-6, (ii) IL-6, Cancer, (iii) IL-6, rs1800795, (iv) IL-6, rs1800796, (v) IL-6, rs1800797, (vi) IL-6 -174G/C or -572G/C or -597G/A, (vii) polymorphisms, (viii) GWAS, (ix) case-control study.

### Eligibility criteria

Two authors independently investigated the title and abstract for all papers and primarily removed the irrelevant and incomplete studies. For the final review the following inclusion-exclusion criteria were used: if the study was (i) designed to measures the association between the IL-6 gene polymorphisms (rs1800795, rs1800796, rs1800797) and cancer risk, (ii) case-control design and (iii) sufficient to provide necessary information of genotypic frequency, it selected for this meta-analysis.

### Data extraction

From the eligible studies several information was compiled for each selected study such as first author, year of the study, country of origin, ethnicity of the study subject, number of case-control, types of cancer, allelic and genotypic distribution and so on. To test the validity of any selected studies for this meta-analysis, Hardy-Weinberg equilibrium (HWE) test was performed using the Chi-square statistic. A selected study was considered as a good study for meta-analysis if $\Pr\{\chi2_{obs} \leq \chi2\} \geq .05$ (Table 1).

### Statistical modeling for meta-analysis

Meta-analysis is a collection of statistical methods to compile the results of similar independent studies. It is used to take the overall decision across a number of similar studies. Let us now introduce the statistical methods that are used in this paper for taking the overall decision about the relationship between the IL-6 gene polymorphisms and cancer risk. At first we have checked the quality of existing studies by testing the Hardy-Weinberg equilibrium (HWE). The HWE test is performed using the Chi-square statistic with the null hypothesis that the

**Table 1. Characteristic of eligible studies included in meta-analysis of the IL-6 gene (rs1800795, rs1800796, rs1800797) polymorphisms.**

| Study | Year | Country | Ethnicity | Cancer | Case/Control | HWE |
|---|---|---|---|---|---|---|
| **rs1800795** | | | | | | |
| Zheng et al. [105] | 2000 | Sweden | Caucasian | Skin Cancer | 73/128 | 0.357(Y) |
| El-Omar et al. [113] | 2003 | USA | Mixed | Stomach Cancer | 213/209 | 0.913(Y) |
| Hwang (b) et al. [114] | 2003 | USA | Caucasian | Stomach Cancer | 30/30 | 0.399(Y) |
| Hwang (a) et al. [114] | 2003 | USA | Asian | Stomach Cancer | 30/30 | 1.000(Y) |
| Howell et al. [104] | 2003 | UK | Caucasian | Skin Cancer | 161/224 | 0.258(Y) |
| Landi et al. [55] | 2003 | France | Caucasian | Colon Cancer | 361/311 | 0.761(Y) |
| Sun et al. [98] | 2004 | USA | Caucasian | Prostate Cancer | 1337/753 | 0.492(Y) |
| Bushley et al. [118] | 2004 | USA | Mixed | Ovarian Cancer | 182/218 | 0.020(N) |
| Campa et al. [81] | 2004 | France | Caucasian | Lung Cancer | 243/207 | 0.818(Y) |
| Smith et al. [34] | 2004 | UK | Caucasian | Breast Cancer | 144/224 | 0.258(Y) |
| Zhang et al. [105] | 2004 | China | Caucasian | Skin Cancer | 241/260 | 0.993(Y) |
| Campa et al. [79] | 2005 | France | Caucasian | Lung Cancer | 1995/1982 | 0.448(Y) |
| Seifart et al. [80] | 2005 | Germany | Caucasian | Lung Cancer | 182/243 | 0.163(Y) |
| Migita et al. [71] | 2005 | Japan | Asian | Liver Cancer | 48/188 | 1.000(Y) |
| Hefler et al. [36] | 2005 | Austria | Caucasian | Breast Cancer | 269/227 | 0.935(Y) |
| Snoussi et al. [33] | 2005 | Tunisia | African | Breast Cancer | 305/200 | 0.829(Y) |
| Leibovici et al. [88] | 2005 | USA | Caucasian | Prostate Cancer | 444/443 | 0.000(N) |
| Festa et al. [103] | 2005 | Sweden | Caucasian | Skin Cancer | 241/260 | 0.993(Y) |
| Cordano et al. [23] | 2005 | UK | Caucasian | Blood Cancer | 408/349 | 0.167(Y) |
| Basturk et al. [123] | 2005 | Turkey | Caucasian | Renal cell | 25/49 | 0.007(N) |
| Mazur et al. [25] | 2005 | Poland | Caucasian | Blood Cancer | 54/50 | 0.239(Y) |
| Kamangar et al. [115] | 2006 | Finland | Caucasian | Stomach Cancer | 102/152 | 0.004(N) |
| Xing et al. [111] | 2006 | China | Asian | Stomach Cancer | 65/71 | 0.141(Y) |
| Michaud et al. [97] | 2006 | USA | Caucasian | Prostate Cancer | 484/613 | 0.832(Y) |
| Vairaktaris et al. [101] | 2006 | Greece | Caucasian | Oral Cancer | 162/156 | 0.298(Y) |
| Cozen et al. [21] | 2006 | USA | Caucasian | Blood Cancer | 146/125 | 0.333(Y) |
| Gunter et al. [61] | 2006 | USA | Caucasian | Colon Cancer | 204/190 | 0.385(Y) |
| Theodoropoulos et al. [54] | 2006 | Greece | Caucasian | Colon Cancer | 222/200 | 0.055(Y) |
| Noguetra et al. [45] | 2006 | Brazil | Mixed | Cervical Cancer | 56/253 | 0.001(N) |
| Balasubramanian et al. [39] | 2006 | UK | Caucasian | Breast Cancer | 497/490 | 0.759(Y) |
| Gonzalez-Zuloeta et al. [40] | 2006 | Netherland | Caucasian | Breast Cancer | 171/3651 | 0.290(Y) |
| Lan et al. [22] | 2006 | USA | Caucasian | Blood Cancer | 510/590 | 0.358(Y) |
| Rothman et al. [24] | 2006 | USA | Caucasian | Blood Cancer | 3066/3499 | 0.506(Y) |
| Slattery et al. [62] | 2007 | USA | Caucasian | Colon Cancer | 1579/1977 | 0.015(N) |
| Slattery et al. [27] | 2007 | USA | Caucasian | Breast Cancer | 650/678 | 0.122(Y) |
| Deans et al. [107] | 2007 | UK | Caucasian | Stomach Cancer | 197/224 | 0.258(Y) |
| Gatti et al. [108] | 2007 | Brazil | Mixed | Stomach Cancer | 56/112 | 0.509(Y) |
| Duch et al. [19] | 2007 | Brazil | Mixed | Blood Cancer | 52/60 | 0.442(Y) |
| Vishnoi et al. [70] | 2007 | India | Asian | Liver Cancer | 124/200 | 0.936(Y) |
| Gonullu et al. [32] | 2007 | Turkey | Caucasian | Breast Cancer | 38/24 | 0.000(N) |
| Vogel et al. [38] | 2007 | Denmark | Caucasian | Breast Cancer | 361/361 | 0.728(Y) |
| Nearman et al. [20] | 2007 | USA | Caucasian | Blood Cancer | 28/362 | 0.120(Y) |
| Crusius et al. [111] | 2008 | France | Caucasian | Stomach Cancer | 243/1138 | 0.044(N) |
| Kesarwani et al. [96] | 2008 | India | Asian | Prostate Cancer | 200/200 | 0.100(Y) |
| Vairaktaris et al. [100] | 2008 | Greece | Caucasian | Oral Cancer | 162/156 | 0.000(N) |
| Colakogullari et al. [78] | 2008 | Turkey | Caucasian | Lung Cancer | 44/58 | 0.221(Y) |

*(Continued)*

**Table 1.** (Continued)

| Study | Year | Country | Ethnicity | Cancer | Case/Control | HWE |
|---|---|---|---|---|---|---|
| **rs1800795** | | | | | | |
| Upadhyay et al. [67] | 2008 | India | Asian | Liver Cancer | 168/201 | 0.586(Y) |
| Kury et al. [60] | 2008 | France | Caucasian | Colon Cancer | 1023/1121 | 0.079(Y) |
| Wilkening et al. [53] | 2008 | Germany | Caucasian | Colon Cancer | 303/580 | 0.481(Y) |
| Ennas et al. [18] | 2008 | Italy | Caucasian | Blood Cancer | 39/112 | 0.506(Y) |
| Slattery et al. [51] | 2009 | USA | Caucasian | Colon Cancer | 750/1250 | 0.016(N) |
| Slattery et al. [52] | 2009 | USA | Caucasian | Colon Cancer | 1839/2014 | 0.015(N) |
| Gangwar et al. [41] | 2009 | India | Asian | Cervical Cancer | 160/200 | 0.371(Y) |
| Ozgen et al. [117] | 2009 | Turkey | Caucasian | Thyroid Cancer | 42/340 | 0.009(N) |
| Moore et al. [93] | 2009 | USA | Caucasian | Prostate Cancer | 957/847 | 0.152(Y) |
| Pierce et al. [94] | 2009 | USA | Caucasian | Prostate Cancer | 175/1934 | 0.132(Y) |
| Wang et al. [95] | 2009 | USA | Caucasian | Prostate Cancer | 253/280 | 0.448(Y) |
| Zabaleta et al. [96] | 2009 | USA | Caucasian | Prostate Cancer | 74/401 | 0.000(N) |
| Talar-Wojnarowska et al. [119] | 2009 | Poland | Caucasian | Pancreatic Cancer | 97/50 | 0.191(Y) |
| Aladzsity et al. [16] | 2009 | Hungary | Caucasian | Blood Cancer | 97/99 | 0.101(Y) |
| Falleti et al. [66] | 2009 | Italy | Caucasian | Liver Cancer | 219/236 | 0.536(Y) |
| Ognjanovic et al. [69] | 2009 | USA | Mixed | Liver Cancer | 117/221 | 0.000(N) |
| Tsilidis et al. [58] | 2009 | USA | Caucasian | Colon Cancer | 203/367 | 0.537(Y) |
| Vasku et al. [59] | 2009 | Czech Republic | Caucasian | Colon Cancer | 100/100 | 0.601(Y) |
| Cherel et al. [30] | 2009 | France | Caucasian | Breast Cancer | 293/82 | 0.695(Y) |
| DeMichele et al. [31] | 2009 | USA | Caucasian | Breast Cancer | 339/100 | 0.569(Y) |
| Andrie et al. [17] | 2009 | Greece | Caucasian | Blood Cancer | 81/81 | 0.777(Y) |
| Ognjanovic et al. [50] | 2010 | USA | Mixed | Colon Cancer | 271/539 | 0.000(N) |
| Zhao et al. [112] | 2010 | China | Asian | Stomach Cancer | 142/200 | 0.943(Y) |
| Dossus et al. [87] | 2010 | Germany | Mixed | Prostate Cancer | 7937/8508 | 0.035(N) |
| Cacev et al. [56] | 2010 | Croatia | Caucasian | Colon Cancer | 160/160 | 0.582(Y) |
| Hawken et al. [63] | 2010 | Canada | Caucasian | Colon Cancer | 1133/1125 | 0.461(Y) |
| Abuli et al. [49] | 2011 | Spain | Caucasian | Colon Cancer | 1416/1424 | 0.672(Y) |
| Grimm et al. [46] | 2011 | Austria | Caucasian | Cervical Cancer | 131/208 | 0.990(Y) |
| Gaur et al. [99] | 2011 | India | Caucasian | Oral Cancer | 140/200 | 0.069(Y) |
| Giannitrapani et al. [65] | 2011 | Italy | Caucasian | Liver Cancer | 105/95 | 0.402(Y) |
| Lima junior et al. [48] | 2012 | Brazil | Mixed | Cervical Cancer | 345/345 | 0.093(Y) |
| Pooja et al. [28] | 2012 | India | Asian | Breast Cancer | 200/200 | 0.000(N) |
| Totaro et al. [120] | 2013 | Italy | Caucasian | Neuroblastoma | 326/511 | 0.646(Y) |
| Pohjnen et al. [109] | 2013 | Finland | Caucasian | Stomach Cancer | 56/179 | 0.706(Y) |
| Chen et al. [72] | 2013 | China | Asian | Lung Cancer | 1237/1252 | 0.903(Y) |
| Bai et al. [73] | 2013 | China | Asian | Lung Cancer | 193/210 | 0.145(Y) |
| Oduor et al. [13] | 2014 | Kenya | African | Blood Cancer | 117/88 | 1.000(Y) |
| Mandal et al. [85] | 2014 | USA | Caucasian | Prostate Cancer | 164/140 | 0.001(N) |
| Gu et al. [14] | 2014 | China | Asian | Blood Cancer | 157/435 | 0.159(Y) |
| Cao et al. [111] | 2014 | China | Asian | Stomach Cancer | 162/162 | 0.210(Y) |
| Shi et al. [43] | 2014 | China | Asian | Cervical Cancer | 518/518 | 0.349(Y) |
| Cil et al. [116] | 2014 | Turkey | Caucasian | Thyroid Cancer | 190/216 | 0.722(Y) |
| Slattery et al. [29] | 2014 | USA | Mixed | Breast Cancer | 3567/4157 | 0.000(N) |
| Chen et al. [89] | 2015 | China | Asian | Prostate Cancer | 212/236 | 0.267(Y) |
| Talaat et al. [15] | 2015 | Egypt | Mixed | Blood Cancer | 100/119 | 0.568(Y) |
| Sampaio et al. [110] | 2015 | Portugal | Caucasian | Stomach Cancer | 50/50 | 0.608(Y) |

*(Continued)*

**Table 1.** (Continued)

| Study | Year | Country | Ethnicity | Cancer | Case/Control | HWE |
|---|---|---|---|---|---|---|
| **rs1800795** | | | | | | |
| Sa-Nguanraksa et al. [37] | 2016 | Thailand | Asian | Breast Cancer | 391/79 | 0.000(N) |
| Pu et al. [42] | 2016 | China | Asian | Cervical Cancer | 360/728 | 0.310(Y) |
| Abana et al. [26] | 2017 | USA | Caucasian | Breast Cancer | 277/711 | 0.490(Y) |
| Winchester et al. [84] | 2017 | USA | Caucasian | Prostate Cancer | 625/532 | 0.169(Y) |
| Attar et al. [106] | 2017 | Iran | Mixed | Stomach Cancer | 100/361 | 0.000(N) |
| Sabrina et al. [44] | 2017 | Tunisia | African | Cervical Cancer | 112/164 | 0.002(N) |
| DargahiAbbasabad et al. [86] | 2018 | Iran | Mixed | Prostate Cancer | 112/250 | 0.000(N) |
| Zhao et al. [121] | 2018 | China | Asian | Neuroblastoma | 130/50 | 0.585(Y) |
| Dos Santos et al. [112] | 2018 | Brazil | Mixed | Stomach Cancer | 52/87 | 0.517(Y) |
| Taheri et al. [86] | 2018 | Iran | Mixed | Prostate Cancer | 130/200 | 0.194(Y) |
| Shuwei Wang et al. [64] | 2018 | China | Asian | Colon Cancer | 186/200 | 0.160(Y) |
| **rs1800796** | | | | | | |
| Hwang (a) et al. [114] | 2003 | USA | Caucasian | Stomach Cancer | 30/30 | 0.020(N) |
| Hwang (b) et al. [114] | 2003 | USA | Asian | Stomach Cancer | 30/30 | 0.394(Y) |
| Sun et al. [98] | 2004 | USA | Caucasian | Prostate Cancer | 1337/753 | 0.211(Y) |
| Xing et al. [111] | 2006 | China | Asian | Stomach Cancer | 65/71 | 0.141(Y) |
| Seow et al. [75] | 2006 | Singapore | Asian | Lung Cancer | 124/162 | 0.560(Y) |
| Kamanger et al. [116] | 2006 | USA | Caucasian | Stomach Cancer | 102/152 | 0.004(N) |
| Slattery et al. [62] | 2007 | USA | Caucasian | Colon Cancer | 1573/1972 | 0.015(N) |
| Bao et al. [91] | 2008 | China | Asian | Prostate Cancer | 136/120 | 0.000(N) |
| Slattery et al. [51] | 2009 | USA | Caucasian | Colon Cancer | 750/1205 | 0.000(N) |
| Kang et al. [111] | 2009 | Korea | Asian | Stomach Cancer | 332/326 | 0.078(Y) |
| Pierce et al. [93] | 2009 | USA | Caucasian | Prostate Cancer | 175/1934 | 0.161(Y) |
| Wang et al. [94] | 2009 | USA | Caucasian | Prostate Cancer | 253/280 | 0.405(Y) |
| Tsilidis et al. [58] | 2009 | USA | Caucasian | Colon Cancer | 203/367 | 0.019(N) |
| Su et al. [82] | 2010 | China | Asian | Lung Cancer | 363/370 | 0.298(Y) |
| Lim et al. [83] | 2011 | Singapore | Asian | Lung Cancer | 298/718 | 0.250(Y) |
| Bai et al. [73] | 2013 | China | Asian | Lung Cancer | 193/210 | 0.145(Y) |
| Chen et al. [72] | 2013 | China | Asian | Lung Cancer | 615/638 | 0.990(Y) |
| Liang et al. [74] | 2013 | China | Asian | Lung Cancer | 138/138 | 0.625(Y) |
| Kiyohara et al. [77] | 2014 | Japan | Asian | Lung Cancer | 462/379 | 0.919(Y) |
| Cao et al. [111] | 2014 | China | Asian | Stomach Cancer | 162/162 | 0.210(Y) |
| Tang et al. [68] | 2014 | China | Asian | Liver Cancer | 505/395 | 0.474(Y) |
| Chen et al. [89] | 2015 | China | Asian | Prostate Cancer | 212/236 | 0.851(Y) |
| Haung et al. [90] | 2016 | China | Asian | Prostate Cancer | 236/256 | 0.094(Y) |
| Zhang et al. [111] | 2017 | China | Asian | Stomach Cancer | 473/474 | 0.750(Y) |
| Xie et al. [111] | 2017 | China | Asian | Stomach Cancer | 400/400 | 0.859(Y) |
| Zhu et al. [35] | 2017 | China | Asian | Breast Cancer | 1514/1540 | 0.204(Y) |
| Dos Santos et al. [112] | 2018 | Brazil | Mixed | Stomach Cancer | 52/87 | 0.555(Y) |
| **rs1800797** | | | | | | |
| Hwang et al. [114] | 2003 | USA | Caucasian | Stomach Cancer | 30/30 | 0.399(Y) |
| Snoussi et al. [33] | 2005 | Tunisia | African | Breast Cancer | 305/200 | 0.830(Y) |
| Festa et al. [102] | 2005 | Sweden | Caucasian | Skin Cancer | 241/260 | 0.385(Y) |
| Rothman et al. [24] | 2006 | USA | Caucasian | Blood Cancer | 2658/3068 | 0.124(Y) |
| Castro et al. [46] | 2009 | Sweden | Caucasian | Cervical Cancer | 973/1763 | 0.584(Y) |
| Pierce et al. [93] | 2009 | USA | Caucasian | Prostate Cancer | 175/1934 | 0.437(Y) |

*(Continued)*

**Table 1.** (Continued)

| Study | Year | Country | Ethnicity | Cancer | Case/Control | HWE |
|---|---|---|---|---|---|---|
| **rs1800795** | | | | | | |
| Tsilidis et al. [58] | 2009 | USA | Caucasian | Colon Cancer | 203/362 | 0.931(Y) |
| Vasku et al. [89] | 2009 | Czech Republic | Caucasian | Colon Cancer | 100/100 | 0.661(Y) |
| DeMichele et al. [31] | 2009 | USA | Caucasian | Breast Cancer | 339/100 | 0.316(Y) |
| Gu et al. [14] | 2014 | China | Asian | Blood Cancer | 93/204 | 0.831(Y) |
| Sa-Nguanraksa et al. [37] | 2016 | Thailand | Asian | Breast Cancer | 391/79 | 0.863(Y) |
| Leng et al. [76] | 2016 | USA | Caucasian | Lung Cancer | 242/336 | 0.346(Y) |
| Sabrina et al. [44] | 2017 | Tunisia | African | Cervical Cancer | 112/164 | 0.000(N) |
| Winchester et al. [84] | 2017 | USA | Caucasian | Prostate Cancer | 625/532 | 0.075(Y) |
| Huang et al. [57] | 2018 | USA | Caucasian | Colon Cancer | 135/269 | 0.745(Y) |
| Dos Santos et al. [112] | 2018 | Brazil | Mixed | Stomach Cancer | 52/87 | 0.446(Y) |

HWE: Hardy-Weinberg equilibrium; Y: Yes; N: No; All the included studies are ordered by the year of publication.

genotypic ratio is consistent for the control population of all studies. The chi-square statistic for this test is given by:

$$\chi^2 = \sum_{i=1}^{3} \frac{(O_i - E_i)^2}{E_i} \qquad (1)$$

which follows chi-square distribution with 1 degree freedom. Here $O_i$ and $E_i$ represents observe and expected frequency of the genotype, respectively. If $p$ and $q$ are the probabilities of $C$ and $G$ allele, respectively and $O_i = obs(i)$ is observe frequency of $i$th genotype among the 3 genotypes $CC$, $CG$ and $GG$. Then $p$ is calculated as:

$$p = \frac{2 \times obs(CC) + obs(CG)}{2 \times (obs(CC) + (obs(CG) + obs(GC))}; \quad \text{and } q = 1 - p \qquad (2)$$

The expected frequency of $i$th genotype is denoted by $E_i = E(i)$ defined as $E(CC) = p^2n$, $E(CG) = 2pqn$, $E(GG) = q^2n$, where $n$ is the total number of observation.

The heterogeneity of different studies has been examined by using Cochran's $Q$ statistic and its extended Higgin's & Thompson $I^2$ statistic [131, 132]. The Cochran's $Q$ statistic is defined as:

$$Q = \sum_{k=1}^{K} w_k \left( \hat{\theta}_k - \frac{\sum_{k=1}^{K} w_k \hat{\theta}_k}{\sum_{k=1}^{K} w_k} \right)^2, \qquad (3)$$

which follows the chi-square distribution with $K$-1 degrees of freedom. Here $\hat{\theta}_k = \ln(\mathrm{OR}_k)$ for the $k$th study, and $w_k = \frac{1}{\hat{\sigma}_k^2}$ is the weight of $k$th study. The variance of the $k$th study can be calculated as:

$$\hat{\sigma}_k^2 = var\left(\ln\left(OR_k\right)\right) = \frac{1}{m_{1k}} + \frac{1}{m_{2k}} + \frac{1}{m_{3k}} + \frac{1}{m_{4k}} \qquad (4)$$

where $m_{1k}$ and $m_{2k}$ indicates the number of exposures and $m_{3k}$ and $m_{4k}$ indicates non-exposures, in the case-control groups of $k$th study, respectively (that is, for the genetic model $C$ vs. $G$, the allele $C$ is exposer and $G$ is non-exposer). The Higgin's & Thompson $I^2$ statistic is

defined as:

$$I^2 = \max\left\{0, \frac{Q - (K - 1)}{Q} \times 100\%\right\} \tag{5}$$

The values of $I^2$ greater than 25%, 50% and 75% indicates the low, moderate, and high heterogeneity among the individual studies, respectively.

The pooled odds ratio (OR) has been applied for checking the significant association between the IL-6 gene polymorphisms and cancer risk under different genetic models like as dominant models [CC + CG vs. GG or AA + AG vs. GG], homozygote models [CC vs. GG or AA vs. GG], over-dominant models [CG vs. CC + GG or AG vs. AA + GG], recessive models [CC vs. CG + GG or AA vs. AG + GG], and allelic contrast models [C vs. G or A vs. G]. To calculate pooled OR for each genetic combination, we have used the random effect model if the $Q$-test suggests the highly significant heterogeneity ($p$-value < 0.10) among different studies; otherwise, fixed effect model are used. We have also estimated 95% confidence interval (CI) of OR based on $Z$-statistic [131, 132]. The OR for the $k$th study is calculated as:

$$OR_k = \frac{\frac{m_{1k}}{m_{2k}}}{\frac{m_{3k}}{m_{4k}}} = \frac{m_{1k}m_{4k}}{m_{2k}m_{3k}}, \tag{6}$$

For the fixed effect model, overall OR is calculated by using the Mentel—Haenszel (M-H) method as follows:

$$\hat{\theta}_k = \theta_F + \epsilon_k; \quad \text{where } \epsilon_k \sim N(0, \hat{\sigma}_k^2) \tag{7}$$

where $\hat{\theta}_F = \widehat{OR}_{MH} = \dfrac{\sum_{k=1}^{K}\left(\frac{m_{1k}m_{4k}}{N_k}\right)}{\sum_{k=1}^{K}\left(\frac{m_{2k}m_{3k}}{N_k}\right)}$

$$= \sum_{k=1}^{K}\left(\frac{\frac{m_{2k}m_{3k}}{N_k}}{\sum_{i=1}^{K}\left(\frac{m_{2k}m_{3k}}{N_k}\right)}\right) \times OR_k, \tag{8}$$

and $N_k = m_{1k} + m_{2k} + m_{3k} + m_{4k}$, and the variance and 95% C.I. of overall effect can be defined as:

$$Var\left(\hat{\theta}_F\right) \frac{1}{\sum_{k=1}^{K}\left(\frac{m_{2k}m_{3k}}{N_k}\right)}; \hat{\theta}_F \pm 1.96\sqrt{Var\left(\hat{\theta}_F\right)}, \tag{9}$$

For the random effect model, overall OR is calculated by using the inverse variance method as follows:

$$\hat{\theta}_k = \theta_R + \nu_k + \epsilon_k; \quad \text{where, } \nu_k \sim N(0, \tau^2) \tag{10}$$

The random parameter $\theta_R$ is calculated as,

$$\hat{\theta}_R = \frac{\sum_{k=1}^{K} w_{kR}\hat{\theta}_k}{\sum_{k=1}^{K} w_{kR}}, \tag{11}$$

Where

$$se(\hat{\theta}_R) = \sqrt{var(\hat{\theta}_R)} = \sqrt{\frac{1}{\sum_{k=1}^{K} w_{kR}}};$$

$$w_{kR} = \frac{1}{\sigma_k^2 + \tau^2}, \quad \text{and} \quad \tau^2 = \frac{Q - (K-1)}{\sum w_k - \left(\frac{\sum w_k^2}{\sum w_k}\right)} \quad (12)$$

However, *Q*-test cannot give the assurance of model adequacy. Therefore we also considered the goodness of fit test to check the model adequacy. To check the model adequacy, we performed three distinct goodness of fit (GoF) tests proposed by Chen et al. [133]. These three GoF tests known as Anderson-Darling (AD) test [134, 135], Cramer-von Mises (CvM) test [135–137] and Shapiro-Wilk (SW) test [138] for testing the null hypothesis that the individual effects follow the normal distribution. If individual effects are significantly normal, then random effect model is used for estimating the combined effect else fixed effect model is used. The test statistic of each normality test is defined as:

$$AD = -K - \sum_{k=1}^{K} \left(\frac{2k-1}{k}\right) \left[\ln F(\hat{\theta}_k) + \ln\left(1 - F(\hat{\theta}_{K+1-k})\right)\right], \quad (13)$$

$$CvM = \frac{1}{12K} + \sum_{k=1}^{K} \left[\frac{2k-1}{2K} - F(\hat{\theta}_k)\right]^2, \quad (14)$$

$$SW = \frac{\left(\sum_{k=1}^{K} a_k \hat{\theta}_k\right)^2}{\sum_{k=1}^{K} (\hat{\theta}_k - \bar{\theta})^2}, \quad (15)$$

where, $\hat{\theta}_k$ is the ordered data, $\bar{\theta}$ is sample mean of $\hat{\theta}_k$, $K$ is sample size means number of individual study, $F(\hat{\theta}_k)$ is cumulative distribution function of normal distribution with $k$th order statistic, $a_k$ is constants generated from means, variances, and covariances of the order statistics. To perform these three tests, Chen et al. [133] proposed the following steps:

**Step 1.** Compute $ad_0$, $cvm_0$, and $sw_0$ from AD, CvM, and SW statistics, respectively, for given $\hat{\theta}_k = \ln(OR_k)$, $k = 1, 2, \ldots, K$;

**Step 2.** Resample B = $10^5$ sub-samples from $MVN(0, \hat{\Sigma})$, where,

$$\hat{\Sigma} = \begin{bmatrix} \hat{\sigma}_1^2 + \hat{\tau}^2 & 0 & \cdots & 0 \\ 0 & \hat{\sigma}_2^2 + \hat{\tau}^2 & \cdots & 0 \\ \vdots & \vdots & \ddots & \vdots \\ 0 & 0 & \cdots & \hat{\sigma}_K^2 + \hat{\tau}^2 \end{bmatrix}$$

Then, compute $ad_j$, $cvm_j$, and $sw_j$ by using AD, CvM, and SW statistics, respectively, for each sample $j$ ($j = 1, 2, \ldots, B$).

**Step 3.** Compute *p*-values by using $\sum_{j=1}^{B} I_{[ad_j > ad_0]}/B$, $\sum_{j=1}^{B} I_{[cvm_j > cvm_0]}/B$ and

$\sum_{j=1}^{B} I_{[sw_j < sw_0]}/B$ for the above three tests, respectively, where $I_{[s > s_0]} = \begin{cases} 1 \text{ for } s > s_0; \\ 0, \text{ otherewise.} \end{cases}$

Then the respective *z*-score is calculated as follows:

$$Z = \begin{cases} \dfrac{\sum_k w_k \hat{\theta}_k}{\sqrt{\sum_k w_k}}, & \text{for fixed effect model} \\[2em] \dfrac{\sum_k w_{kR} \hat{\theta}_k}{\sqrt{\sum_k w_{kR}}}, & \text{for random effect model} \end{cases} \tag{16}$$

Subgroup analyses are also executed based on ethnicity and type of cancer by using the techniques mentioned above.

We have performed the sensitivity analysis using the full data and the reduced data that are obtained by removing the studies those are failed to pass the HWE validation and publication bias test. The publication bias is examined for each study visually by funnel plot and significantly by Egger regression test [139] and Begg's test [140]. The Egger regression test statistic is defined as:

$$T = \frac{\hat{a}}{se(\hat{a})} \tag{17}$$

which follows the *t*-distribution with ($K$-2) degrees of freedom under the null hypothesis $H_0$: $\alpha$ = 0 (no publication bias), $\hat{\alpha}$ is obtained by the least square estimation using one of the following models:

$$\hat{\theta}_k \sqrt{w_k} = \alpha + \mu \sqrt{w_k} + \varepsilon_k, \text{ for fixed effect model, and} \tag{18}$$

$$\hat{\theta}_k \sqrt{w_{kR}} = \alpha + \mu \sqrt{w_{kR}} + \varepsilon_k, \text{ for random effect model,} \tag{19}$$

where $\varepsilon_k \sim iid\ N(0, \sigma^2)$. The Begg's test statistic is defined as:

$$Z = \frac{C - D}{\sqrt{K(K-1)(2K+5)/18}}, \tag{20}$$

which follows asymptotically $N(0,1)$ under the null hypothesis $H_0$: $\alpha$ = 0 (no publication bias). Here $C$ and $D$ are the number of concordant and discordant, respectively, those are obtained by using the Kendall's ranking of $t_k^*$ and $\hat{\sigma}_k^2$ or $\hat{\sigma}_{kR}^2$. Here:

$$t_k^* = \frac{t_k - \bar{t}}{\sqrt{\vartheta_k^*}} \tag{21}$$

where, $t_k = \mathrm{OR}_k$ is the OR of $k$th study, and:

$$\bar{t} = \begin{cases} \dfrac{\sum_k w_k t_k}{\sqrt{\sum_k w_k}}, & \text{for fixed effect model} \\[4ex] \dfrac{\sum_k w_{kR} t_k}{\sqrt{\sum_k w_{kR}}}, & \text{for random effect model} \end{cases} \tag{22}$$

$$\vartheta_k^* = \begin{cases} \hat{\sigma}_k^2 - \dfrac{1}{\sum w_k}, & \text{for fixed effect model} \\[4ex] \hat{\sigma}_{kR}^2 - \dfrac{1}{\sum w_{kR}}, & \text{for random effect model} \end{cases} \tag{23}$$

We have used the 'meta' R-package (http://meta-analysis-with-r.org/) for implementing the above statistical methods for the meta-analysis.

## Results

### Study characteristics

In this meta-analysis, first we reviewed 580 articles which mentioned the IL-6 gene in their titles and abstracts. Then 477 articles were selected after removing the duplication. Again we removed 337 articles due to the absence of full text, case-control and cancer related studies. Finally 118 articles were selected for final review by removing some studies having incomplete information. The flow chart of the studies selection process was shown in Fig 1. The finally selected articles included 103 studies for the rs1800795 SNP with 45238 cases and 57255 controls (Table 2), 27 studies for the rs1800796 SNP with 10733 cases and 13405 controls (Table 3), 16 studies for the rs1800797 SNP with 6674 cases and 9493 controls (Table 4). These articles were classified to different types of cancer such as blood cancer, breast cancer, cervical cancer, colon cancer, liver cancer, lung cancer, neuroblastoma, oral cancer, ovarian cancer, pancreatic cancer, prostate cancer, skin cancer, stomach cancer and thyroid cancer. For being the single study, ovarian cancer, renal cell carcinoma (RCC) and pancreatic cancer for the rs1800795

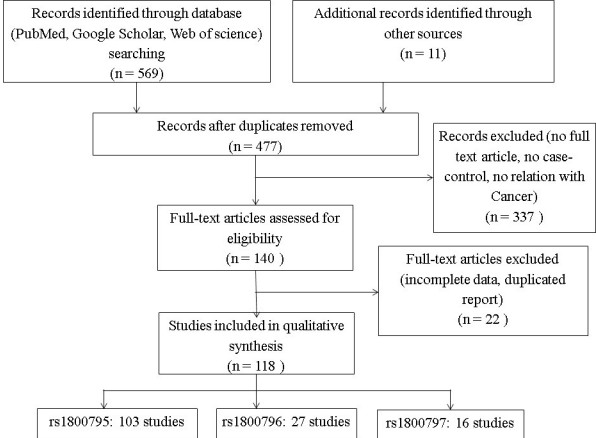

**Fig 1. Flow diagram of study selection for the IL-6 gene polymorphisms; where 'n' is the number of studies.**

SNP, breast cancer and liver cancer for rs1800796 SNP and, lung cancer and skin cancer for the rs1800797 SNP were organized in a subgroup entitled as other cancer (Tables 2–4).

## Quantitative synthesis

**IL-6 rs1800795 SNP.**   In the overall analysis, we found that the rs1800795 SNP was not associated with overall risk of cancer under five genetic models [C vs. G: OR = 1.02, 95% CI = 0.97–1.06, p-value = 0.445; CC vs. GG: OR = 1.06, 95% CI = 0.98–1.16, p-value = 0.1429; CC vs. CG + GG: OR = 1.05, 95% CI = 0.98–1.12, p-value = 0.2054; CC + CG vs. GG: OR = 1.02, 95% CI = 0.99–1.05, p-value = 0.0615; CG vs. CC + GG: OR = 0.99, 95% CI = 0.96–1.01, p-value = 0.2689] (Table 2 and S1A–S1E Fig in S1 File).

The subgroup analysis through the types of cancer showed the significant association that the IL-6 -174G/C polymorphism performed a protective role in liver cancer for four genetic models [C vs. G: OR = 0.73, 95% CI = 0.62–0.86, p-value = 0.0002; CC vs. GG: OR = 0.61, 95% CI = 0.42–0.88, p-value = 0.0082; CC vs. CG + GG: OR = 0.68, 95% CI = 0.48–0.97, p-value = 0.0321; CC + CG vs. GG: OR = 0.69, 95% CI = 0.56–0.85, p-value = 0.0004]; increased the risk for cervical cancer under four genetic models [C vs. G: OR = 1.29, 95% CI = 1.07–1.56, p-value = 0.0075; CC vs. GG: OR = 1.63, 95% CI = 1.06–2.52, p-value = 0.0266; CC vs. CG + GG: OR = 1.57, 95% CI = 1.25–1.97, p-value = 0.0001; CC + CG vs. GG: OR = 1.31, 95% CI = 1.05–1.64, p-value = 0.0178]; as well as increasing

**Table 2.  Meta-analysis of the IL-6 -174G/C polymorphism association with cancer risk.**

| | Study Number | Sample Size | CC vs. GG | | CC vs. CG + GG | | CC + CG vs. GG | | CG vs. CC + GG | | C vs. G | |
|---|---|---|---|---|---|---|---|---|---|---|---|---|
| | | | OR[a] (95% CI[b]) | *p*-value | OR (95% CI) | *p*-value | OR (95% CI) | *p*-value | OR (95% CI) | *p*-value | OR (95% CI) | *p*-value |
| **Overall** | 103 | 102493 | 1.06 [0.98; 1.16] | 0.1429 | 1.05 [0.98; 1.12] | 0.2054 | 1.02 [0.99; 1.05] | 0.0615 | 0.99 [0.96; 1.01] | 0.2681 | 1.02 [0.97; 1.06] | 0.4459 |
| **Blood Cancer** | 13 | 10824 | 1.04 [0.86; 1.26] | 0.6246 | 1.06 [0.86; 1.29] | 0.6289 | 0.98 [0.87; 1.10] | 0.7399 | 0.99 [0.90; 1.08] | 0.7508 | 1.01 [0.96; 1.07] | 0.8411 |
| **Breast Cancer** | 14 | 18686 | 0.93 [0.84; 1.02] | 0.1329 | 0.96 [0.88; 1.05] | 0.4042 | 0.95 [0.82; 1.11] | 0.5291 | 0.98 [0.91; 1.04] | 0.4804 | 1.00 [0.90; 1.11] | 0.9619 |
| **Cervical Cancer** | 7 | 4098 | **1.63 [1.06; 2.52]** | **0.0266** | **1.57 [1.25; 1.97]** | **0.0001** | **1.31 [1.05; 1.64]** | **0.0178** | 1.14 [0.89; 1.47] | 0.2927 | **1.29 [1.07; 1.56]** | **0.0075** |
| **Colon Cancer** | 15 | 21263 | 0.98 [0.85; 1.13] | 0.8097 | 0.98 [0.87; 1.08] | 0.4136 | 1.02 [0.92; 1.14] | 0.6893 | 1.02 [0.97; 1.08] | 0.4196 | 1.01 [0.93; 1.09] | 0.7632 |
| **Liver Cancer** | 6 | 1922 | **0.61 [0.42; 0.88]** | **0.0082** | **0.68 [0.48; 0.97]** | **0.0321** | **0.69 [0.56; 0.85]** | **0.0004** | 0.75 [0.54; 1.05] | 0.0908 | **0.73 [0.62; 0.86]** | **0.0002** |
| **Lung Cancer** | 6 | 7846 | 1.08 [0.85; 1.36] | 0.5296 | 0.97 [0.76; 1.23] | 0.7981 | 1.08 [0.86; 1.84] | 0.5137 | 1.07 [0.86; 1.34] | 0.5334 | 1.01 [0.87; 1.18] | 0.9102 |
| **Neuroblastoma** | 2 | 1017 | 1.18 [0.71; 1.95] | 0.5244 | 1.15 [0.71; 1.88] | 0.5731 | 1.06 [0.84; 1.81] | 0.6865 | 1.05 [0.80; 1.37] | 0.7259 | 1.08 [0.88; 1.33] | 0.4570 |
| **Oral Cancer** | 3 | 896 | 0.69 [0.08; 6.27] | 0.7430 | 0.74 [0.18; 3.04] | 0.6714 | 0.80 [0.16; 4.09] | 0.7864 | 0.86 [0.25; 2.95] | 0.8160 | 0.81 [0.25; 2.58] | 0.7197 |
| **Prostate Cancer** | 14 | 28441 | **1.24 [1.05; 1.46]** | **0.0096** | **1.19 [1.03; 1.39]** | **0.0202** | 1.09 [0.98; 1.22] | 0.1142 | 0.97 [0.91; 1.05] | 0.4773 | **1.09 [1.00; 1.19]** | **0.0441** |
| **Skin Cancer** | 4 | 1588 | 0.99 [0.75; 1.32] | 0.968 | 0.96 [0.76; 1.20] | 0.6564 | 1.05 [0.84; 1.31] | 0.6808 | 1.08 [0.88; 1.31] | 0.4597 | 1.00 [0.87; 1.16] | 0.9912 |
| **Stomach Cancer** | 14 | 4503 | 1.12 [0.85; 1.50] | 0.4087 | 1.07 [0.81; 1.40] | 0.6405 | 1.08 [0.92; 1.22] | 0.4317 | 1.04 [0.90; 1.19] | 0.6024 | 1.04 [0.94; 1.16] | 0.4517 |
| **Thyroid Cancer** | 2 | 788 | 1.20 [0.67; 2.13] | 0.5364 | 1.30 [0.75; 2.27] | 0.3477 | 0.87 [0.63; 1.21] | 0.4149 | 0.79 [0.57; 1.10] | 0.1680 | 0.97 [0.76; 1.25] | 0.8399 |
| **Other Cancers** | 3 | 621 | 1.10 [0.58; 2.10] | 0.7683 | 1.09 [0.73; 1.61] | 0.6802 | 1.29 [0.80; 2.13] | 0.3236 | 1.07 [0.74; 1.56] | 0.7148 | 1.14 [0.85; 1.54] | 0.3711 |
| **Ethnicity[C]** | | | | | | | | | | | | |
| **African** | 3 | 986 | 1.58 [0.68; 3.66] | 0.2895 | 1.40 [0.61; 3.25] | 0.4248 | **1.66 [1.20; 2.3]** | **0.0027** | **1.64 [1.17; 2.30]** | **0.0037** | **1.54 [1.16; 2.04]** | **0.0030** |
| **Asian** | 19 | 10043 | **1.56 [1.19; 2.03]** | **0.0011** | **1.37 [1.22; 1.53]** | **0.0001** | **1.17 [1.08; 1.29]** | **0.0024** | 0.93 [0.85; 1.02] | 0.1151 | **1.20 [1.12; 1.29]** | **0.00001** |
| **Caucasian** | 66 | 62535 | 1.01[0.92; 1.11] | 0.8193 | 1.00 [0.94; 1.07] | 0.9469 | 1.00 [0.97; 1.04] | 0.8788 | 1.00 [0.97; 1.04] | 0.9164 | 1.00 [0.95; 1.05] | 0.9291 |
| **Mixed** | 15 | 28929 | 0.97 [0.74; 1.27] | 0.8224 | 1.02 [0.80; 1.31] | 0.8629 | 0.89 [0.77; 1.04] | 0.1493 | 0.89 [0.78; 1.03] | 0.1134 | 0.93 [0.82; 1.06] | 0.2676 |

Statistical significance presented in Bold.

[a]Odds Ratio;

[b]Confidence Interval ORs for the ethnicity subgroups are for overall cancer risk.

**Table 3. Meta-analysis of the IL-6 -572G/C polymorphism association with cancer risk.**

| | Study Number | Sample size | CC vs. GG | | CC vs. CG + GG | | CC + CG vs. GG | | CG vs. CC + GG | | C vs. G | |
|---|---|---|---|---|---|---|---|---|---|---|---|---|
| | | | OR[a] (95% CI[b]) | p-value | OR (95% CI) | p-value | OR (95% CI) | p-value | OR (95% CI) | p-value | OR (95% CI) | p-value |
| Overall | 27 | 24138 | 1.03 [0.85; 1.25] | 0.7635 | 0.99 [0.86; 1.14] | 0.8582 | 1.07 [0.94; 1.22] | 0.2931 | **1.12 [1.01; 1.23]** | **0.0288** | 1.04 [0.95; 1.15] | 0.3839 |
| Colon Cancer | 3 | 6070 | 1.04 [0.67; 1.64] | 0.8507 | 1.05 [0.73; 1.50] | 0.8000 | 1.04 [0.67; 1.63] | 0.8507 | 1.07 [0.85; 1.36] | 0.5552 | 1.10 [0.79; 1.53] | 0.5613 |
| Lung Cancer | 7 | 4808 | 1.13 [0.75; 1.69] | 0.5575 | 0.93 [0.74; 1.17] | 0.5400 | **1.31 [1.04; 1.65]** | **0.0228** | **1.31 [1.08; 1.59]** | **0.0072** | 1.19 [0.98; 1.43] | 0.0734 |
| Prostate Cancer | 6 | 5928 | **0.52 [0.37; 0.72]** | **0.0001** | **0.67 [0.53; 0.84]** | **0.0005** | **0.74 [0.61; 0.90]** | **0.0025** | 1.00 [0.84; 1.18] | 0.9811 | **0.74 [0.64; 0.85]** | **0.0000** |
| Stomach Cancer | 9 | 3378 | **1.41 [1.10; 1.81]** | **0.0076** | **1.29 [1.07; 1.55]** | **0.0080** | **1.41 [1.09; 1.81]** | **0.0088** | 1.02 [0.79; 1.31] | 0.8940 | **1.16 [1.03; 1.30]** | **0.0069** |
| Other Cancer | 2 | 3954 | 1.13 [0.47; 2.68] | 0.7852 | 0.90 [0.78; 1.04] | 0.1400 | 1.03 [0.64; 1.67] | 0.8922 | 1.12 [0.99; 1.28] | 0.0841 | 1.05 [0.72; 1.53] | 0.7878 |
| Ethnicity[C] | | | | | | | | | | | | |
| Asian | 18 | 12883 | 1.02 [0.79; 1.31] | 0.8812 | 1.00 [0.83; 1.20] | 0.9636 | 1.06 [0.91; 1.25] | 0.4557 | **1.13 [1.01; 1.27]** | **0.0293** | 1.04 [0.92; 1.19] | 0.4974 |
| Caucasian | 8 | 11116 | 1.07 [0.76; 1.49] | 0.7105 | 0.97 [0.78; 1.21] | 0.7964 | 1.10 [0.87; 1.39] | 0.4398 | 1.12 [0.89; 1.40] | 0.3391 | 1.04 [0.87; 1.26] | 0.6445 |
| Mixed | 1 | 139 | 3.20 [.28; 36.45] | 0.3487 | 3.44 [.30; 38.90] | 0.3181 | 0.83 [0.37; 1.86] | 0.6590 | 0.70 [0.30; 1.63] | 0.4130 | 0.97 [0.48; 1.98] | 0.9379 |

Statistical significance presented in Bold.

[a]Odds Ratio;

[b]Confidence Interval;

[C] ORs for the ethnicity subgroups are for overall cancer risk.

the risk for prostate cancer [C vs. G: OR = 1.09, 95% CI = 1.00–1.19, p-value = 0.0441; CC vs. GG: OR = 1.24, 95% CI = 1.05–1.46, p-value = 0.0096; CC vs. CG + GG: OR = 1.19, 95%

**Table 4. Meta-analysis of the IL-6 -597G/A polymorphism association with cancer risk.**

| | Study Number | Sample size | AA vs. GG | | AA vs. AG + GG | | AA + AG vs. GG | | AG vs. AA + GG | | A vs. G | |
|---|---|---|---|---|---|---|---|---|---|---|---|---|
| | | | OR[a] (95% CI[b]) | p-value | OR (95% CI) | p-value | OR (95% CI) | p-value | OR (95% CI) | p-value | OR (95% CI) | p-value |
| Overall | 16 | 16167 | 0.96 [0.85; 1.08] | 0.5152 | 0.97 [0.87; 1.07] | 0.5064 | 1.00 [0.93; 1.08] | 0.9289 | 0.98 [0.91; 1.05] | 0.5025 | 0.99 [0.94; 1.04] | 0.7169 |
| Blood Cancer | 2 | 6023 | 0.97 [0.83; 1.13] | 0.7353 | 0.90 [0.84; 1.12] | 0.6699 | 1.01 [0.47; 6.86] | 0.8045 | 0.97 [0.88; 1.07] | 0.5815 | 1.75 [0.48; 6.46] | 0.3981 |
| Breast Cancer | 3 | 1414 | 1.11 [0.63; 1.92] | 0.7097 | 1.07 [0.64; 1.79] | 0.7980 | 1.24 [0.82; 1.88] | 0.0800 | 0.79 [0.59; 1.06] | 0.1176 | 1.20 [0.95; 1.52] | 0.1221 |
| Cervical Cancer | 2 | 3012 | **0.79 [0.63; 0.98]** | **0.0390** | **0.82 [0.68; 1.00]** | **0.0474** | 0.93 [0.75; 1.23] | 0.3969 | 0.94 [0.80; 1.09] | 0.4312 | 0.91 [0.82; 1.02] | 0.0667 |
| Colon Cancer | 3 | 1174 | 0.84 [0.60; 1.20] | 0.3674 | 0.86 [0.64; 1.18] | 0.3700 | 0.93 [0.72; 1.20] | 0.5885 | 0.97 [0.77; 1.24] | 0.8488 | 0.93 [0.78; 1.10] | 0.3887 |
| Prostate Cancer | 2 | 3266 | 0.97 [0.71; 1.31] | 0.8243 | 1.07 [0.81; 1.42] | 0.6183 | 0.87 [0.72; 1.05] | 0.1708 | 1.17 [0.97; 1.41] | 0.0944 | 0.95 [0.83; 1.09] | 0.4591 |
| Stomach Cancer | 2 | 199 | 1.71 [0.54; 5.34] | 0.3551 | 1.83 [0.60; 5.57] | 0.2846 | 0.98 [0.54; 1.76] | 0.9657 | 1.21 [0.67; 2.20] | 0.5301 | 1.10 [0.69; 1.76] | 0.6853 |
| Other Cancers | 2 | 1079 | 1.36 [0.96; 1.92] | 0.0744 | 1.20 [0.90; 1.60] | 0.2122 | 1.29 [0.96; 1.73] | 0.0524 | 0.91 [0.71; 1.16] | 0.4556 | **1.19 [1.00; 1.41]** | **0.0450** |
| Ethnicity[C] | | | | | | | | | | | | |
| African | 2 | 781 | 0.80 [0.37; 1.70] | 0.5600 | 0.69 [0.32; 1.46] | 0.3323 | **1.48 [1.08; 2.03]** | **0.0135** | **0.61 [0.44; 0.84]** | **0.001** | **1.28 [0.98; 1.67]** | **0.0191** |
| Asian | 2 | 767 | 0.61 [.02; 15.10] | 0.7600 | 0.62 [0.02; 15.24] | 0.7669 | 2.11 [0.91; 4.85] | 0.0602 | 0.56 [0.11; 2.89] | 0.0800 | 2.11 [0.93; 4.81] | 0.0753 |
| Caucasian | 11 | 14480 | 0.97 [0.85; 1.10] | 0.5965 | 0.97 [0.87; 1.08] | 0.5689 | 0.98 [0.91; 1.05] | 0.5254 | 1.00 [0.93; 1.07] | 0.9640 | 0.97 [.87; 1.08] | 0.5689 |
| Mixed | 1 | 139 | 1.29 [0.36; 4.67] | 0.6900 | 1.44 [0.42; 4.96] | 0.5672 | 0.86 [0.43; 1.72] | 0.6774 | 1.30 [0.65; 2.62] | 0.4600 | 0.98 [0.57; 1.04] | 0.9343 |

Statistical significance presented in Bold.

[a]Odds Ratio;

[b]Confidence Interval;

[C] ORs for the ethnicity subgroups are for overall cancer risk.

CI = 1.03–1.39, p-value = 0.0202]. The blood cancer, breast cancer, colon cancer, lung cancer, neuroblastoma, oral cancer, skin cancer, stomach cancer, thyroid cancer, ovarian cancer and pancreatic cancer showed insignificant associations with the IL-6 -174G/C polymorphism (Table 2).

The subgroup analysis according to ethnicity showed that the IL-6 -174G/C polymorphism was not significantly associated with the cancer risk of Caucasian and mixed populations (Table 2). The subgroup analysis showed the significant association with the increasing overall cancer risk of Asian population under four genetic models [C vs. G: OR = 1.20, 95% CI = 1.12–1.29, p-value = 0.0000; CC vs. GG: OR = 1.56, 95% CI = 1.08–2.03, p-value = 0.0011; CC vs. CG + GG: OR = 1.37, 95% CI = 1.22–1.53, p-value = 0.0001; CC + CG vs. GG: OR = 1.17, 95% CI = 1.08–1.29, p-value = 0.0024] and African population [C vs. G: OR = 1.54, 95% CI = 1.16–2.04, p-value = 0.0030; CC + CG vs. GG: OR = 1.66, 95% CI = 1.20–2.30, p-value = 0.0027; CG vs. CC + GG: OR = 1.64, 95% CI = 1.17–2.30, p-value = 0.0037] (Table 2).

**Source of heterogeneity.** We observed significant heterogeneity in the analysis of the IL-6 rs1800795 (-174G /C) polymorphism for overall cancer [CC vs.GG: Q = 258.44, df = 97, p-value = 0.0001, $\tau^2$ = 0.0747, $I^2$ = 62.47%; CC vs. CG + GG: Q = 228.48, df = 97, p-value = 97, $\tau^2$ = .0478, $I^2$ = 57.53%; CC + CG vs. GG: Q = 333.19, df = 100, p-value = 0.0001, $\tau^2$ = .0485, $I^2$ = 69.98%; CG vs. CC + GG: Q = 297.07, df = 100, p-value = 0.0001, $\tau^2$ = .0385, $I^2$ = 66.34%; C vs. G: Q = 378.15, df = 100, p-value = 0.0001, $\tau^2$ = .0290, $I^2$ = 73.56%]. The subgroup analysis corresponding to cancer type and ethnicity were performed to observe the sources of heterogeneity. The results of our analysis suggested that the studies in breast cancer, cervical cancer, colon cancer, lung cancer, oral cancer, prostate cancer, stomach cancer, and the ethnicity of Asian, Caucasian and Mixed population were the main sources of heterogeneity (S1 Table).

**IL-6 rs1800796 SNP.** The results generated through this meta-analysis showed that the IL-6 -572G/C polymorphism was significantly associated with the overall cancer risk in the case of over-dominant model [CG vs. CC + GG: OR = 1.12, 95% CI = 1.01–1.23, p-value = 0.0288] (Table 3 and S1F–S1J Fig in S1 File). Though, it was not significantly associated with the overall cancer risk under the other four genetic models (Allelic, dominant, recessive and homozygote).

The subgroup analysis through the types of cancer showed the significant association that the IL-6 rs1800796 (-572G/C) performed a protective role in prostate cancer for four genetic models [C vs. G: OR = 0.74, 95% CI = 0.64–0.85, p-value = 0.0000; CC vs. GG: OR = 0.52, 95% CI = 0.37–0.72, p-value = 0.0001; CC vs. CG + GG: OR = 0.67, 95% CI = 0.53–0.84, p-value = 0.0005; CC + CG vs. GG: OR = 0.74, 95% CI = 0.61–0.90, p-value = 0.0025]. The IL-6 -572G/C polymorphism was also exhibited significant association with the increasing risk of stomach cancer under four genetic models [C vs. G: OR = 1.16, 95% CI = 1.03–1.30, p-value = 0.0069; CC vs. GG: OR = 1.41, 95% CI = 1.10–1.81, p-value = 0.0076; CC vs. CG + GG: OR = 1.29, 95% CI = 1.07–1.55, p-value = 0.0080; CC + CG vs. GG: OR = 1.41, 95% CI = 1.09–0.81, p-value = 0.0088] and lung cancer for genetic models [CC + CG vs. GG: OR = 1.31, 95% CI = 1.04–1.65, p-value = 0.0228; CG vs. CC + GG: OR = 1.31, 95% CI = 1.08–1.59, p-value = 0.0072]. The colon, breast and liver cancers showed insignificant association with this polymorphism (Table 3).

The subgroup analysis based on ethnicity, the Asian population suggested that the IL-6 -572G/C polymorphism was significantly associated with increasing overall cancer risk for the over-dominant model [CG vs. CC + GG: OR = 1.13, 95% CI = 1.01–1.27, p-value = 0.0293]. The Caucasian and mixed ethnic group showed insignificant association of the IL-6 -572G/C polymorphism with the overall cancer risk (Table 3).

**Source of heterogeneity.** We found the significant heterogeneity of different studies in the analysis of IL-6 -572G/C polymorphism for overall cancer risk under the all genetic models

[C vs. G: Q = 89.96, df = 26, p-value = 0.0001, $\tau^2$ = .0391, $I^2$ = 71.04; CC vs. GG: Q = 55.82, df = 26, p-value = 0.0006, $\tau^2$ = .1033, $I^2$ = 53.41%; CC vs. CG + GG: Q = 49.23, df = 26, p-value = .0039, $\tau^2$ = .0459, $I^2$ = 47.18%; CC + CG vs. GG: Q = 76.19, df = 26, p-value = .0001, $\tau^2$ = .0618, $I^2$ = 65.88%; CG vs. CC + GG: Q = 54.40, df = 26, p-value = .0009, $\tau^2$ = .0293, $I^2$ = 52.21%]. We also explored the sources of heterogeneity by the subgroup analysis based on cancer type and ethnic group. The results of our analysis suggested that the colon, lung, breast and liver cancers with the ethnic group of Asian and Caucasian were the main sources of heterogeneity of different studies (S1 Table).

**IL-6 rs1800797 SNP.**   The finding of our analysis suggested that the IL-6 rs1800797 (-597G/A) polymorphism were not significantly associated with overall cancer risk under genetic models [A vs. G: OR = 0.99, 95% CI = 0.94–1.04, p-value = 0.7169; AA vs.GG: OR = 0.96, 95% CI = 0.85–1.08, p-value = 0.5152; AA + AG vs. GG: OR = 1.00, 95% CI = 0.93–1.08, p-value = 0.9289; AA vs. AG + GG: OR = 0.97, 95% CI = 0.87–1.07, p-value = 0.5064; AG vs. AA + GG: OR = 0.98, 95% CI = 0.91–1.05, p-value = 0.5025] (Table 4 and S1K–S1O Fig in S1 File).

The subgroup analysis based on cancer type showed that the blood, breast, colon, prostate and stomach cancers were not significantly associated with the IL-6 -597G/A polymorphism (Table 4). It also showed the significant role of IL-6 -597G/A polymorphism with the decreasing of cervical cancer risk under some genetic models [AA vs. GG: OR = 0.79, 95% CI = 0.63–0.98, p-value = 0.0390; AA vs. AG + GG: OR = 0.82, 95% CI = 0.68–1.00, p-value = 0.0474] and increasing of lung and skin cancer risks under the allelic model [A vs. G: OR = 1.19, 95% CI = 1.00–1.41, p-value = 0.0450] (Table 4).

The subgroup analysis based on ethnicity, the Asian, Caucasian and mixed population suggested that the IL-6 rs1800797 (-597G/A) polymorphism was not significantly associated with the overall cancer risk. Only for African population showed the significant association between this polymorphism and overall cancer risk by three genetic models [A vs. G: OR = 1.28, 95% CI = 0.98–1.67, p-value = 0.0191; AA + AG vs. GG: OR = 1.48, 95% CI = 1.08–2.03, p-value = 0.0135; AG vs. AA + GG: OR = 0.61, 95% CI = 0.44–0.84, p-value = 0.0010] (Table 4).

**Source of heterogeneity.**   We found the insignificant heterogeneity of different studies in the analysis of IL-6 -597G/A polymorphism for overall cancer risk under the all genetic models. The subgroup analysis corresponding to cancer type and ethnic group were performed to observe the sources of heterogeneity. We found that only blood cancer was the main source of heterogeneity [A vs. G: Q = 6.56, df = 1, p-value = 0.0104, $\tau^2$ = 0.7679, $I^2$ = 84.80%; AA + AG vs. GG: Q = 6.66, df = 1, p-value = 0.0099, $\tau^2$ = 0.8110, $I^2$ = 85.00%] (S1 Table).

## Publication bias

In this study the funnel plot was used to check the publication bias of IL-6 -174G/C and IL-6 -572G/C polymorphisms with the allelic model C versus G and IL-6 -597G/A polymorphism with the allelic model A versus G. According to the funnel plot, the distribution of ORs in terms of standard errors (SEs) was symmetric for each of three polymorphisms (-174G/C, -572G/C, -597G/A) and no publication bias was observed among the selected studies for this meta-analysis (Fig 2). Also, publication bias was checked through performing Begg's test and Egger's linear regression test. Results generated through both the Egger's and Begg's tests also suggested that there is no significant publication bias for the polymorphisms with the genetic models [C vs. G: p-value = 0.4778 (0.8030), and CC vs. GG: p-value = 0.5667 (0.7403) for the rs1800795 SNP; C vs. G: p-value = 0.3267 (0.6022) and CC vs. GG: p-value = 0.1664 (0.2347) for the rs1800796 SNP; A vs. G: p-value = 0.1175 (0.1768) and AA vs. GG: p-value = 0.6016

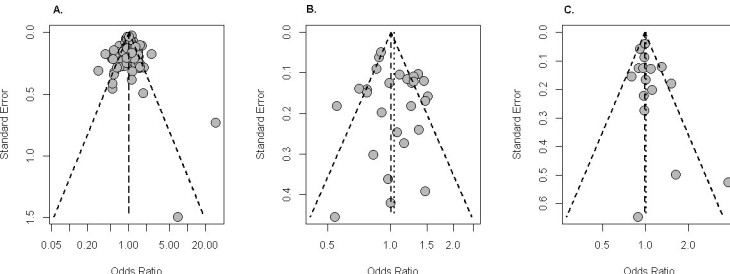

**Fig 2. Funnel plot of the IL-6 polymorphisms to showing visual evidence of no publication bias.** (A) -174G/C for C vs. G, (B) -572G/C for C vs. G and (C) -597G/A for A vs. G.

(0.7290) for the rs1800797 SNP] (see also S2A and S2B Table in S2 File). The p-value inside the first parenthesis was obtained by the Begg's test.

## Sensitivity analysis

The sensitivity analysis was conducted to increase the reliability of this meta-analysis. First, the meta-analysis was conducted considering all studies. Then, the studies that did not pass the HWE test were removed and the meta-analysis was performed again using the reduced dataset of the respective genetic models. The analyzed results showed an insignificant change of association which suggested that the meta-analysis analysis data generated through this study is both stable and robust (see S2C–S2E Table in S2 File).

## Discussion and conclusion

In this paper we discussed the way of statistical modeling for meta-data analyses in details incorporating the goodness of fit test for checking the model adequacy. Then multi-case meta-analysis was conducted to find out the association of cancer risk with each of three SNPs (rs1800795, rs1800796, rs1800797) of the IL-6 gene. A total of 118 individual studies which included 50053 case and 65204 control samples, based on different cancers and ethnic groups were included in this extensive meta- analysis. The results computed through this study suggested that the IL-6 rs1800795 polymorphism is insignificantly associated with the overall cancer risk, but significantly reduced the risk of liver cancer under four genetic models (CC vs. GG; CC vs. CG + GG; CC + CG vs. GG; CG vs. CC + GG; C vs. G), which is in line with the previously reported multi-case meta-analysis in [130]. Also, this SNP showed significant association with the increasing risk of cervical and prostate cancers, where the results of cervical cancer are supported by the previous single-case meta-analysis in [123], but not with the multi-case meta-analysis in [130]. The results calculated for IL-6 rs1800796 polymorphism also showed significant association with overall cancer risk for one genetic model. This polymorphism showed significant association with the prostate and stomach cancers under four genetic models (CC vs. GG; CC vs. CG + GG; CC + CG vs. GG; C vs. G), where these results are supported by the previous multi-case meta-analysis in [130] and single-case meta-analysis in [111], respectively. Moreover, the results generated through this meta-analysis indicated that the rs1800796 polymorphism is significantly associated with the increasing risk of lung cancer. The IL-6 rs1800797 polymorphism analyzed data showed insignificant association with cancer risk, which is supported by previous single-case meta-analysis in [125]. Also, the results of this study showed the significant association of IL-6 rs1800797 polymorphism with increasing risk of cervical cancer, which showed insignificant association in [130].

The ethnicity based subgroup analysis data showed significant association between the rs1800795 polymorphism and the overall cancer risk of both African under three genetic models and Asian populations under four genetic models(CC vs. GG; CC vs. CG + GG; CC + CG vs. GG; C vs. G). For rs1800796 polymorphisms results suggested the significant association with the cancer risk of Asian populations. Also, the rs1800797 polymorphism was significantly associated with African ethnic groups for the cancer risk. All the results of subgroup analysis by ethnicity were supported by the previous multi-case meta-analysis in [130]. Thus, we observed that our multi-case meta-analysis results received more support than the previous multi-case meta-analysis results in [130] from the other single-case meta-analysis results in [123–129].

It should be mentioned here again that all of the previous meta-analyses [123–130] did not check the model adequacy through the goodness of fit test. To estimate the combined effects, all of them used fixed effect (FE) or random effect (RE) models based on Cochran's homogeneity test though the sample sizes were small for some individual studies. For being small sample sizes, the individual effects may not be followed the normal distribution and the Cochran's test may be produced misleading results about the homogeneity of individual effects. However, in our case, we used the GoF test suggested by Chen et al. [133] to fix the lack of model fitting. We observed that some of our fitted models contradict with the fitted models based on Cochran's homogeneity test and significant changes in association between gene polymorphisms and cancer risks. In particularly, we observed the changes with some overall and subgroup cases of all polymorphisms (rs1800795, rs1800796, rs1800797). Due to the contradictory model selections, contradictory associations were also observed for three cases of rs1800795 polymorphism (liver cancer: CG vs. CC + GG; Asian ethnicity: CC + CG vs.GG and C vs. G) and single case of the rs1800796 polymorphism (stomach cancer: CC + CG vs. GG). However, there were some limitations on conducting this meta-analysis like for heterogeneity factors such as age, sex, family history, levels of IL-6 expression were not considered and that might affect the association. The literature reviewed and selected for this study was in English language only; therefore, the publication bias could not be completely avoided or some selection bias might occur. Also, the small sample size may affect the results for some types of cancer.

In conclusion, the results of this study indicated that the IL-6 gene is significantly associated with the overall cancer risk. Particularly, this gene showed significant association with 5 types of cancer risks (liver, prostate, cervical, stomach and lung) and insignificant association with 11 types of cancer risks (blood, breast, colon, neuroblastoma, oral, skin, thyroid, ovarian, pancreatic and renal cell carcinoma) by the sub-group analysis of cancer types. Comparative discussion showed that our current multi-case meta-analysis results received more support than any other individual previous meta-analysis results about the association between the IL-6 gene SNPs (rs1800795, rs1800796 and rs1800797) and different types of cancer risks. Therefore, the results generated through this detailed systematic meta-analysis based on larger sample size of the IL-6 gene polymorphisms provides more evidence for further exploring the IL-6 gene as a very potent prognostic biomarker for early detection of various types of cancers.

## Supporting information

**S1 Table. Heterogeneity analysis of IL-6 gene polymorphisms.**
(DOCX)

**S1 File. Forest plot of IL-6 gene polymorphisms (rs1800795, rs1800796, rs1800797) for five genetic models.**
(DOCX)

**S2 File. Egger's linear regression and Begg's test of IL-6 gene polymorphisms for checking the publication bias.**
(DOCX)

**S3 File. PRISMA checklist.**
(DOC)

**S1 Data. Full dataset of IL-6 gene rs1800795 polymorphism.**
(XLSX)

**S2 Data. Full dataset of IL-6 gene rs1800796 polymorphism.**
(XLSX)

**S3 Data. Full dataset of IL-6 gene rs1800797 polymorphism.**
(XLSX)

## Acknowledgments

We are grateful to the editor and reviewers for their valuable comments that help us to improve the manuscript.

## Author Contributions

**Conceptualization:** Jesmin, Md. Nurul Haque Mollah.

**Data curation:** Md. Harun-Or-Roshid, Md. Borqat Ali, Jesmin.

**Formal analysis:** Md. Harun-Or-Roshid, Md. Borqat Ali, Jesmin, Md. Nurul Haque Mollah.

**Funding acquisition:** Md. Harun-Or-Roshid, Md. Nurul Haque Mollah.

**Investigation:** Md. Harun-Or-Roshid.

**Methodology:** Md. Harun-Or-Roshid, Md. Nurul Haque Mollah.

**Project administration:** Md. Nurul Haque Mollah.

**Resources:** Md. Harun-Or-Roshid, Jesmin.

**Software:** Md. Harun-Or-Roshid.

**Supervision:** Md. Nurul Haque Mollah.

**Writing – original draft:** Md. Harun-Or-Roshid.

**Writing – review & editing:** Jesmin, Md. Nurul Haque Mollah.

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
