## [Editor Report · Decision Letter 0]

21 Sep 2020

PONE-D-20-23035

Statistical meta-analysis to investigate the association between interleukin-6 gene polymorphisms and cancer risk

PLOS ONE

Dear Dr. Mollah,

Thank you for submitting your manuscript to PLOS ONE. After careful consideration, we feel that it has merit but does not fully meet PLOS ONE’s publication criteria as it currently stands. Therefore, we invite you to submit a revised version of the manuscript that addresses the points raised during the review process.

In the manuscript, the authors performed meta-analyses with large sample sizes. The findings, if based on appropriate statistical methods, from this study may provide us useful information in the area. However, when you apply "standard" meta-analysis models (e.g., fixed and random effect models), you need to check the fit of the models; otherwise, the outputs from inadequate models might be misleading. Please perform goodness of fit tests for your models. You can use the gof tests described in the following reference: Chen, Z., Zhang, G., & Li, J. (2015). Goodness-of-fit test for meta-analysis. *Scientific reports*, *5*, 16983. If indeed your models are lack of fit, please revise your paper accordingly (e.g., you can follow the suggestions offered in the above mentioned reference). After receiving your revised manuscript, I will be happy to reconsider your manuscript for possible publication.

We look forward to receiving your revised manuscript.

Kind regards,

Zhongxue Chen

Academic Editor

PLOS ONE

Journal Requirements:

2. In your financial disclosure, please clearly specify whether the funders played any role in the study.

State what role the funders took in the study. If the funders had no role in your study, please state: “The funders had no role in study design, data collection and analysis, decision to publish, or preparation of the manuscript.”If any authors received a salary from any of your funders, please state which authors and which funders.

3. Please note that according to our submission guidelines (http://journals.plos.org/plosone/s/submission-guidelines), outmoded terms and potentially stigmatizing labels should be changed to more current, acceptable terminology. For example: “Caucasian” should be changed to “white” or “of [Western] European descent” (as appropriate).
---

## [Author Response · Author response to Decision Letter 0]

6 Nov 2020

Response to Review Comments

Review Comments: Thank you for submitting your manuscript to PLOS ONE. After careful consideration, we feel that it has merit but does not fully meet PLOS ONE’s publication criteria as it currently stands. Therefore, we invite you to submit a revised version of the manuscript that addresses the points raised during the review process. 

In the manuscript, the authors performed meta-analyses with large sample sizes. The findings, if based on appropriate statistical methods, from this study may provide us useful information in the area. However, when you apply "standard" meta-analysis models (e.g., fixed and random effect models), you need to check the fit of the models; otherwise, the outputs from inadequate models might be misleading. Please perform goodness of fit tests for your models. You can use the gof tests described in the following reference: Chen, Z., Zhang, G., & Li, J. (2015). Goodness-of-fit test for meta-analysis. Scientific reports, 5, 16983. If indeed your models are lack of fit, please revise your paper accordingly (e.g., you can follow the suggestions offered in the above mentioned reference). After receiving your revised manuscript, I will be happy to reconsider your manuscript for possible publication.

Author’s response: Thank you very much for your kind suggestion. We revised our manuscript accordingly. At first, we introduced the importance GoF test in the last paragraph of introduction section (page 4, highlighted portion with blue color). Then we provided the short summary of GoF test in the materials and methods section (pages 13-14, highlighted portion with blue color). We observed that in some cases GoF test based fitted models contradicts with the fitted models based on Cochran’s homogeneity test and changes the previous decision about the association between gene polymorphisms and cancer risks. In particularly, we observed the changes with some overall and subgroup cases of all polymorphisms (rs1800795, rs1800796, rs1800797). For example,

• For rs1800795, the contradictory model were observed for overall cases and Caucasian ethnic group under two genetic combinations (CC + CG vs. GG; CG vs. CC + GG); blood, liver, skin and stomach cancer under all genetic combinations (CC vs. GG; CC vs. CG + GG; CC + CG vs. GG; CG vs. CC + GG; C vs. G); breast cancer under three genetic combinations (CC vs. GG; CC vs. CG + GG; CG vs. CC + GG); colon cancer (CC vs. CG + GG), lung cancer (CC vs. GG; CC + CG vs. GG); prostate cancer (CG vs. CC + GG); Asian ethnic group under four genetic combinations (CC vs. CG + GG; CC + CG vs. GG; CG vs. CC + GG; C vs. G) (Please see additional file ‘S1 Table’, Highlighted portion with blue color). The significant association were changed in the case of liver cancer (CG vs. CC + GG) (Table 2, page 13, highlighted portion with blue color) and Asian ethnicity (CC + CG vs. GG and C vs. G) (Table 2, page 17, highlighted portion with bold & blue color).

• For rs1800796, the contradictory model were observed for the case of colon cancer under three genetic combinations (CC vs. CG + GG; CC + CG vs. GG; CG vs. CC + GG); prostate and stomach cancer for all genetic combinations (CC vs. GG; CC vs. CG + GG; CC + CG vs. GG; CG vs. CC + GG; C vs. G); Caucasian populations (CC vs. GG; CC vs. CG + GG) (Please see additional file ‘S1 Table’, Highlighted portion with blue color). The significant association were changed only the case of stomach cancer (CC + CG vs. GG) (Table 3, page-18, highlighted portion with bold & blue color).

• For rs1800797, the contradictory model were observed for the case of overall cancer under two genetic combinations (CC vs. GG; CC vs. CG + GG); and Caucasian populations under all genetic combinations (CC vs. GG; CC vs. CG + GG; CC + CG vs. GG; CG vs. CC + GG; C vs. G) (Please see additional file ‘S1 Table’, Highlighted portion with blue color). There were no significant association changes were observed (Table 3, page-19). 

We also updated the discussion and conclusion section based on the revised manuscript (page 24-25, highlighted with blue color)

Reference: Chen, Z., Zhang, G., & Li, J. (2015). Goodness-of-fit test for meta-analysis., Scientific reports, 5, 16983)

---

## [Decision Letter · Decision Letter 1]

15 Jan 2021

PONE-D-20-23035R1

Statistical meta-analysis to investigate the association between interleukin-6 gene polymorphisms and cancer risk

PLOS ONE

Dear Dr. Mollah,

Thank you for submitting your manuscript to PLOS ONE. After careful consideration, we feel that it has merit but does not fully meet PLOS ONE’s publication criteria as it currently stands. Therefore, we invite you to submit a revised version of the manuscript that addresses the points raised during the review process.

Please pay special attention to the reference list, there were several replicates.

We look forward to receiving your revised manuscript.

Kind regards,

Zhongxue Chen

Academic Editor

PLOS ONE

Reviewers' comments:

Reviewer's Responses to Questions

**Comments to the Author**

1. If the authors have adequately addressed your comments raised in a previous round of review and you feel that this manuscript is now acceptable for publication, you may indicate that here to bypass the “Comments to the Author” section, enter your conflict of interest statement in the “Confidential to Editor” section, and submit your "Accept" recommendation.

Reviewer #1: (No Response)

2. Is the manuscript technically sound, and do the data support the conclusions?

Reviewer #1: Yes

3. Has the statistical analysis been performed appropriately and rigorously? 

Reviewer #1: Yes

4. Have the authors made all data underlying the findings in their manuscript fully available?

Reviewer #1: Yes

5. Is the manuscript presented in an intelligible fashion and written in standard English?

Reviewer #1: No

6. Review Comments to the Author

Reviewer #1: Thank you for the opportunity to review this manuscript. This is a well-conducted meta-analysis that thoroughly analyzed the associations between these three IL6 SNPs and cancer risk. I have a few minor comments:

1. This manuscript could benefit from some proofreading by a native English speaker, as there are a number of grammatical errors throughout the paper. In particular, the phrase "IL-6 gene" needs to have the word "the" preceding it, or it can be replaced by "IL6" in italics without the word "the" to refer to the gene itself.

2. Are the studies listed in Table 1 in any particular order within each SNP? If so, this should be included as a footnote. If not, I would recommend putting them in some order that would make it easier for the reader to review.

3. In Tables 2-4, add footnotes indicating that the ORs for the ethnicity subgroups are for overall cancer risk. Similarly, specify in the methods/results/discussion that the models are estimating overall cancer risk within each ethnicity subgroup.

4. In Table 4, there is a typographical error for the OR for breast cancer in the column for CG vs. CC + GG. There is a space missing between the OR and the CI: 0.98[0.91; 1.04]

7. PLOS authors have the option to publish the peer review history of their article (what does this mean?). If published, this will include your full peer review and any attached files.

Reviewer #1: No

---

## [Author Response · Author response to Decision Letter 1]

28 Jan 2021

Dear Professor Zhongxue Chen

Thank you very much for giving us the opportunity to re-revise our manuscript (ID: PONE-D-20-23035R1) according to the reviewer comments for possible publication in PLOS ONE. We revised our manuscript accordingly as follows: 

Reviewer #1 General Comments: Thank you for the opportunity to review this manuscript. This is a well-conducted meta-analysis that thoroughly analyzed the associations between these three IL6 SNPs and cancer risk. I have a few minor comments:

Minor comments-1: This manuscript could benefit from some proofreading by a native English speaker, as there are a number of grammatical errors throughout the paper. In particular, the phrase "IL-6 gene" needs to have the word "the" preceding it, or it can be replaced by "IL6" in italics without the word "the" to refer to the gene itself.

Author’s response: Thanks for your kind suggestion, we tried to revise the manuscript accordingly and replaced “IL-6 gene” by the “the IL-6 gene” throughout the manuscript (Highlighted by Track changes).

Minor comments-2: Are the studies listed in Table 1 in any particular order within each SNP? If so, this should be included as a footnote. If not, I would recommend putting them in some order that would make it easier for the reader to review.

Author’s response: Yes, the studies listed in Table 1were ordered according to the year of publication for individual SNPs (Table 1, pages 5-8). Also, we added a footnote for the reader’s convenience (Table 1, page 8, Highlighted by Track changes). Thank you so much for the suggestions and going through in that detail. 

Minor comments-3: In Tables 2-4, add footnotes indicating that the ORs for the ethnicity subgroups are for overall cancer risk. Similarly, specify in the methods/results/discussion that the models are estimating overall cancer risk within each ethnicity subgroup.

Author’s response: Thanks again you for your suggestions about Tables 2-4.We updated Tables 2-4 accordingly and added footnote with each table to indicate that the ORs for the ethnicity subgroups are for overall cancer risk (Table 2, page 16; Table 3, page 17; Table 4, page 18; Highlighted by Track changes). We also specified this issue in the results and discussion section of the manuscript (Pages 19, 21-24).

Minor comments-4: In Table 4, there is a typographical error for the OR for breast cancer in the column for CG vs. CC + GG. There is a space missing between the OR and the CI: 0.98[0.91; 1.04]

Author’s response: Thank you so much for your comments. This error was found in Table 2 and we updated it carefully (Table 2, page 16; Highlighted by Track changes).

---

## [Editor Report · Decision Letter 2]

1 Feb 2021

Statistical meta-analysis to investigate the association between the interleukin-6 (IL-6) gene polymorphisms and cancer risk

PONE-D-20-23035R2

Dear Dr. Mollah,

We’re pleased to inform you that your manuscript has been judged scientifically suitable for publication and will be formally accepted for publication once it meets all outstanding technical requirements.

Kind regards,

Zhongxue Chen

Academic Editor

PLOS ONE
---

## [Editor Report · Acceptance letter]

5 Feb 2021

PONE-D-20-23035R2 

Statistical meta-analysis to investigate the association between the Interleukin-6 (IL-6) gene polymorphisms and cancer risk 

Dear Dr. Mollah:

I'm pleased to inform you that your manuscript has been deemed suitable for publication in PLOS ONE. Congratulations! Your manuscript is now with our production department. 

Kind regards, 

on behalf of

Dr. Zhongxue Chen 

Academic Editor

PLOS ONE